# Differential microRNA and Target Gene Expression in Scots Pine (*Pinus sylvestris* L.) Needles in Response to Methyl Jasmonate Treatment

**DOI:** 10.3390/genes16010026

**Published:** 2024-12-27

**Authors:** Baiba Krivmane, Dainis Edgars Ruņģis

**Affiliations:** Latvian State Forest Research Institute “Silava”, 111 Rigas St., LV-2169 Salaspils, Latvia; baiba.krivmane@silava.lv

**Keywords:** conifers, induced resistance, microRNA, defense responses, transcriptome

## Abstract

**Background/objectives:** Methyl jasmonate is a plant signaling molecule involved in a wide range of functions, including stress responses. This study investigates the relative differential expression of microRNAs and their target genes in response to methyl jasmonate treatment of Scots pine needles. **Methods:** A combined strategy of high-throughput sequencing and in silico prediction of potential target genes was implemented. **Results:** a total of 58 differentially expressed (DE) microRNAs (miRNAs) (43 up-regulated and 15 down-regulated), belonging to 29 miRNA families, were identified. The 41 DE miRNAs from 17 families were conifer-specific miRNA families—miR946, miR947, miR950, miR1312, miR1313, miR1314, miR3693, miR3107, miR11452, miR11466, miR11487, miR11490, miR11504, miR11511, miR11532, miR11544, and miR11551. The other DE miRNAs (miR159, miR164, miR169, miR396, miR397, miR398, miR408, miR535) were conserved miRNAs, which are also found in angiosperm species. Transcriptome analysis identified 389 gene transcripts with 562 miRNA-target sites targeted by 57 of the 58 DE miRNAs. Of these, 250 target genes with 138 different GO annotations were found for the 41 DE conifer-specific conserved miRNAs. **Conclusions:** The 26 DE miRNAs from 14 DE miRNA families, of which almost all (12 families, 24 miRNAs) are conifer specific, and were associated with 68 disease resistance and TMV resistance proteins, TIR-NBS-LRR, LRR receptor-like serine/threonine-protein kinase, putative CC-NBS-LRR protein, and putative NBS-LRR protein target transcripts with 29 target gene GO term descriptions. Some of the genes targeted by conifer-specific miRNAs have been previously reported to be targeted by other miRNAs in angiosperms, indicating that the miRNA-target gene regulation system can vary between species.

## 1. Introduction

Conifers of the pine family (*Pinaceae*) dominate forests in the northern hemisphere [1,2,3]. Scots pine (*Pinus sylvestris* L.) is among the most broadly distributed conifer species [4,5] with global ecological, social, and economic importance. Scots pine has a large, diploid genome (2C = 46.02 pg, 22 Gbp) [6,7], which contains numerous pseudogenes, gene families, large inter-gene regions, repetitive sequences transposable elements, and regions representing non-coding and small RNAs (including microRNAs) [8,9,10,11].

MicroRNAs (miRNAs, miRs) are non-coding, single-stranded RNA molecules approximately 22 nucleotides in length, which are effective modifiers of gene expression in plants, regulating gene expression at a post-transcriptional level [12,13]. The miRNAs can inhibit translation or mediate degradation of target messenger RNAs (mRNAs) by complementary base pairing. A single miRNA can bind to many distinct mRNA transcripts and in many cases, one mRNA molecule can contain a number of miRNA binding sites [13]. The miRNAs can bind to target transcripts even if there is not perfect sequence complementarity between the miRNA and target gene sequences, and therefore a relatively small number of miRNAs can simultaneously influence the expression pattern of a larger number of target genes [14].

The control of translation and mRNA degradation is an important part of the regulation of gene expression. Plant miRNAs mainly induce down-regulation of their target genes via degradation of mRNA; however, translational repression has also been described as another mechanism of influencing gene expression [15,16,17,18]. The miRNAs can bind to, and post-transcriptionally influence the expression of, a large number of protein-coding genes, and therefore they have a significant role in the modulation and fine-tuning of a very wide range of biological processes [19].

Jasmonic acid (JA) and its methyl ester, methyl jasmonate (MeJA), are molecules involved in plant signaling processes and have been reported to be involved in a wide range of processes regulating plant growth and development in general as well as being involved in more specific responses, e.g., attacks by pathogens, wounding, and other abiotic and biotic stresses [20,21]. Disruption of JA biosynthesis or changes in signal transduction pathways can induce male sterility [22,23,24,25,26,27] and increase susceptibility to attack by herbivores and pathogens [28,29]. MeJA down-regulates miR156, miR168, miR169, miR172, miR396, miR480, and miR1310 and up-regulates miR164 and miR390 in Chinese yew [30]. MeJA down-regulates 12 and up-regulates 20 miRNA families in Norway spruce bark [31]. Expression of 49 known miRNAs and 15 novel miRNAs was induced in JA-treated wild-type *Arabidopsis thaliana*, whereas expression of one novel miRNA and 22 known miRNAs were repressed in the JA-deficient allene oxide synthase mutant [32]. High-throughput sequencing and bioinformatics tools have been used to investigate the influence of various factors on miRNA and gene expression in plants [33,34].

Scots pine is widespread and, because of its productivity and good wood quality, is an economically significant tree species in northern Europe. MeJA can stimulate induced resistance, a natural defense mechanism in conifers against various biotic and abiotic factors, and increase conifer resistance to pests and pathogens, while also negatively affecting tree growth [35]. Treatment with MeJA directly up-regulates the expression of some defense genes and indirectly influences and modulates the expression of other genes [36]. A previous study of Scots pine [37] reported that MeJA application increased thaumatin-like protein (TLP) expression, although it also had a negative phytotoxic effect, causing the wilting of all MeJA-treated ramets after six weeks. Many of the Norway spruce miRNAs that are differentially expressed after treatment with MeJA are predicted to target receptor genes that are involved in pathogen detection [31]. Therefore, investigation of the molecular mechanisms of gene expression and regulation in response to MeJA treatment can provide additional information about the positive effects of the induction of defense responses in Scots pine, and also about potentially negative effects, such as reduced growth and increased mortality, particularly in young individuals.

The aim of the present study was to investigate the effect of MeJA treatment of Scots pine needles on miRNA and gene expression and to analyze the relative expression of the expressed miRNAs and their target genes. A previous study in our laboratory identified and characterized both conserved and novel miRNAs that were differentially expressed after MeJA treatment of Scots pine needles [38]. In addition, potential precursor sequences and target genes of the identified differentially expressed miRNAs were identified to investigate the range of processes that are regulated in stress conditions by both conserved and novel miRNAs. In the present study, a combined strategy of high-throughput sequencing and in silico prediction of potential target genes and the processes that they are potentially involved in was implemented to investigate the effect of MeJA on *P. sylvestris* miRNAs and the transcriptome. Differentially expressed (DE) miRNAs and target genes in MeJA-treated Scots pine needles are identified and their role is discussed.

## 2. Materials and Methods

### 2.1. Plant Material and Extraction of RNA

The plant material and experimental design were the same as described previously [38]. Briefly, a total of six one-year-old ramets derived from one Scots pine clone (Sm9-III-2 -derived) were analyzed. Two weeks before treatment with MeJA, they were transferred into growth chambers. Three ramets were treated with 5 mL of a 10 mM MeJA/0.1% Tween 80 solution in deionized water, and three control ramets were treated with 5 mL of a 0.1% Tween 80 solution in deionized water. The MeJA-treated and control ramets were grown in separate climate chambers (8 h dark and 16 h light at 17 °C and 22 °C, respectively). Needles were harvested from all ramets two weeks after treatment and kept at –80 °C until extraction of total RNA. Harvested needles (100 mg) for total RNA isolation were ground in liquid nitrogen prior to RNA extraction using a 2% (*w*/*v*) CTAB extraction buffer and a phenol: chloroform: isoamyl alcohol extraction protocol [39]. RNA was isolated from all samples in two replicates from the same ramet—for small RNA and transcriptome sequencing. Contaminating DNA was removed from the total RNA samples using DNase I (Thermofisher Scientific, Waltham, MA, USA), according to the manufacturer’s protocol.

### 2.2. Small RNA Enrichment, Library Preparation and Sequencing Analysis

The isolated total RNA samples were enriched for small RNAs as outlined in the Ion RNA-Seq Library Preparation guide (Thermofisher Scientific Manual 4476286 revision E). The Agilent Technologies 2100 Bioanalyzer with the Agilent Small RNA kit was used to check the quality and quantity. RNA integrity (RIN) values of the samples used in downstream analysis exceeded 8. Six small RNA non-barcoded libraries were made using an Ion Total RNA-Seq Kit v2 for Small RNA Libraries (Part No. 4476289, Thermofisher Scientific). Each amplified small RNA library was quantified, and the quality was analyzed using an Agilent Technologies 2100 Bioanalyzer with a High Sensitivity DNA Kit. Template-positive Ion Sphere™ Particles containing clonally amplified DNA were prepared with the Ion OneTouch™ 2 Instrument using the Ion PGM Template OT2 200 Kit (#4480974) according to the manufacturer’s instructions. The ISP enrichment was then assessed using a Qubit^®^ 2.0 Fluorometer and Ion Sphere™ Quality Control Kit. Enriched ISPs were then loaded onto an Ion 316 chip (Cat. No. 4483188) and sequenced individually on an Ion Personal Genome Machine^®^ System at the Norwegian Institute of Bioeconomy Research according to the manufacturer’s protocol as reported previously [38]. Low-quality reads and adapter sequences were removed and sequences were filtered by length: minimum length 19 nt and maximum length 25 nt, as reported previously [38]. Initial sequencing of the Sm5 library yielded a low number of reads compared to the other libraries, and therefore this library was resequenced. Unique small RNA sequences were deposited to the SRA (Short Read Archive, NCBI) with the accession number PRJNA531446 (https://www.ncbi.nlm.nih.gov/sra/?term=PRJNA531446, URL accessed on 28 November 2024). Data from a total of seven sequencing runs were deposited—MeJA-treated libraries (Sm3, Sm4, Sm8), control libraries (Sm5, Sm5-1 (resequenced), Sm6, Sm7). The identification of conserved and novel miRNAs from this data set has been reported previously [38], and in the present study, differentially expressed miRNAs were identified and analyzed.

### 2.3. Transcriptome Library Preparation and Sequencing Analysis

RNA samples for transcriptome sequencing were enriched for polyadenylated mRNAs using the Dynabeads mRNA DIRECT™ Micro Kit (Life Technologies, Carlsbad, CA, USA). The Agilent Technologies 2100 Bioanalyzer with the Agilent RNA 6000 Nano and Pico Kit was used to check the quality, quantity, and integrity number. The quantity after transcript enrichment of RNA from one of the MeJA-treated samples was not sufficient for library construction; therefore, each of the five transcriptome libraries was produced using 1 μg total RNA with the Ion Total RNA-Seq Kit v2 (Life Technologies) and Ion OneTouch2 System (Thermo Fisher Scientific Inc.) with the Ion PGM Template OT2 200 Kit and sequenced on an Ion Personal Genome Machine^®^ (PGM™) System at the Norwegian Institute of Bioeconomy Research (NIBIO) according to the manufacturer’s instructions and as reported previously [40]. All initial data processing was performed on the Ion Torrent Server version 4.0.2. Initial sequencing of the libraries Sm3, Sm4, and Sm5 yielded a low number of reads compared to the other libraries, and therefore these libraries were resequenced. Transcriptome sequences were deposited to the SRA with the accession number PRJNA1115068 (https://www.ncbi.nlm.nih.gov/sra/PRJNA1115068, URL accessed on 28 November 2024). Data from a total of eight sequencing runs were deposited—MeJA-treated libraries (Sm3, Sm3_repeated, Sm4, and Sm4_repeated), control libraries (Sm5, Sm5_repeated, Sm6, and Sm7). An analysis of the transcriptome and differentially regulated genes using this data set has been reported previously [40]. In the present study, target genes of differentially expressed miRNAs were identified from the transcriptome data.

### 2.4. Data Analysis

Small RNA sequences and transcriptomes were analyzed using the CLC Genomics Workbench software version 7.5.1. Barcode and adapter trimming, quality trimming, short read filtering, read mapping to the reference transcriptome [41], differential gene expression analysis, and transcript annotation (using Blast2GO PRO plugin v. 1.12.11 for the CLC Genomic Workbench software) were conducted as described previously [40,42] using the PS2_trinity Scots pine transcriptome reference [41]. Using the CLC Genomics Workbench software, sequences were counted and assigned to families by comparison with mature miRNA sequences from miRBase. The parameters for sequence comparison were used the same as described previously [38]. Conifer-specific miRNAs were considered to be those that were identified in at least two conifer species.

Potential target genes have been identified by searching for complementary regions between the identified miRNAs in this study and by using all the *Pinus* transcript sequence input and the sequenced *Pinus sylvestris* transcriptome from this study using an online web server and the psRNATarget-Plant Small RNA Target Analysis Server (currently available at https://www.zhaolab.org/psRNATarget/, URL accessed on 28 November 2024) with the maximum expectation value 5, as described previously [43].

Potential target genes were annotated using the CLC Genomic Workbench software Blast2GO PRO Plugin according to GO categories using all non-redundant GeneBank CDS translations + PDB + SwissProt sequences as well SwissProt–non-redundant UniProtKB/SwissProt sequences [44].

MiRNA and gene expression networks were generated using the Cytoscape 3.8.2 program [45].

### 2.5. Statistical Analysis

Statistical analysis for small RNA between control and MeJA samples was conducted using the Empirical Analysis of DGE package in the CLC Genomics Workbench software, which implements the ‘Exact Test’ for two-group comparisons and is incorporated in the EdgeR Bioconductor package version 3.4.0. Each group was analyzed by “*p*-value”, “Fold change” and “Weighted difference” after data normalization. The differential expression analysis of the transcriptome data was conducted using the RNA-Seq tool in the CLC Genomics Workbench software with the following parameters: mismatch cost 2, insertion cost 3, length and a similarity fraction of 0.8, and maximum number of hits for a read of 10. Each group was analyzed by “Max group means”, “-log2 fold change”, “Fold change”, “FDR *p*-value”, and “Bonferroni”.

The differences between the groups (with MeJA treatment and control) were analyzed using CLC Genomics Workbench version 12.0.3. Expressed data were normalized and, using the total normalization method, counts were reported as reads per 1,000,000. The total count filter cut-off was 5.0, and Bonferroni correction was used. After normalization, DE miRNAs between the two compared groups were identified using fold change values ≥1.5 and statistical significance *p*-value ≤0.05. For the transcriptome analysis, fold change values of ≥2.0 and a statistical significance FDR *p*-value of ≤0.05 were used. A *p*-value of ≤0.05 was used as the cut-off criterion for GO enrichment analyses.

## 3. Results

### 3.1. Sequencing of Small RNA Libraries

Sequencing of the six small RNA libraries yielded approximately 5.8 million total reads and 4.5 million reads after trimming (removal of sequences according to length: minimum length 19 nt and maximum length 25 nt). A total of 2,178,900 small RNA reads from control samples and 2,309,559 small RNA reads from samples with MeJA treatment after trimming were identified. The number of small RNAs >19 nt in length was lower in the controls, whereas the number of small RNAs of <19 nt in length increased (Figure 1). Prior to trimming, the average length of small RNA reads in the control sample libraries was 21.57 nt and 20.76 nt in the MeJA-treated sample libraries, but after trimming, the average length was similar in control libraries (21.50 nt) and MeJA-treated libraries (21.46 nt).

A total of 1,021,696 unique small RNA sequences were obtained, 4975 potentially conserved miRNA sequences were identified, annotated, and assigned to 173 miRNA groups, and 1029 potential novel miRNAs that had no homology to miRBase v22 annotations were found, based on the results described previously [38]. The variations of canonical miRNA sequences that were found by nucleotide additions at the 3′ or 5′ end, deletion, or substitution resulting in new sequences different from the annotated miRNA are referred to as isomiRs [46].

A total of 688 miRNA sequences from 55 families were found with an expression level ≥10, including 146 sequences from 33 families with an expression level ≥100 and 42 sequences from 16 families with an expression level ≥1000. The three most highly expressed miRNA families identified in this study were miR951, miR950, and miR946. These three families have previously been reported as being expressed in conifer species (*Pinus sylvestris*, *Pinus taeda* L., *Pinus densata* L, *Picea abies* L.) [31,38,47,48,49] (Appendix A), and also in *Betula pendula* Roth., but with low expression values [50]. Most of the miRNA isomiRs were found in both control and MeJA-treated samples (Appendix A). IsomiRs that were found only in one sample type mostly had very low expression levels—less than 10 transcripts. Only six isomiRs from families miR157 (14 sequences; MeJA), miR159 (13 sequences; MeJA), miR396 (11 sequences; control), miR398 (10 sequences; MeJA), miR946 (11 sequences; MeJA), and miR3701 (18 sequences; MeJA) had an expression level ≥10.

The most highly expressed novel miRNA families were miR00005, miR00017, and miR0009. Four isomiRs: miR00003-028, miR00002-027, miR00002-044, and miR00014-035, were found only in one sample type at an expression level of ≥10 (Appendix A). All of these miRNAs were found in MeJA-treated samples. A total of 119 miRNA sequences from 27 families were found with an expression level ≥10, including 25 sequences from 17 families with an expression level ≥100 and 4 sequences from 3 families with an expression level ≥1000.

### 3.2. Differentially Expressed microRNAs

Of 1,021,696 unique small RNA sequences, 58 miRNAs from 29 families were confirmed to exhibit significant DE levels between the control and MeJA-treated samples (Appendix A) [38].

There were 43 up-regulated in MeJA-treated samples and 15 miRNAs were down-regulated in MeJA-treated samples, based on normalized expression means (Table 1, Appendix A). The length of DE miRNAs varied from 20–24 nt. miRNAs of 21 nt in length were the most prevalent among the obtained sequences (29 from a total of 58 DE sequences) and almost all of these were observed to be up-regulated (26 sequences). Most miRNA families consisted of only one isomiR, except for the miR00024, miR11466, miR11487, miR11504, miR11511, miR159, miR3693, miR398, miR946, miR947, and miR950 families, which contained 2–7 members (Table 1). The 41 DE miRNAs from 17 families were conifer-specific (found in at least two different conifer species or genera) conserved miRNA families, although families miR947, miR950, and miR951 have been found also in *Betula pendula*, but with low expression values—9, 15, 31—and in different maturation stages [50]. Additionally, 11 DE miRNAs from eight families were conserved miRNAs, which are highly conserved among plants [51] or previously have been described in specific taxonomic groups, while another five miRNAs from three families (miR00002, miR00024, and miR00028) have been found only in Scots pine.

In 18 cases, from 58 DE miRNAs, the normalized expression means were lower than 1 (6 in MeJA-treated samples and 12 in control samples), indicating that these miRNAs were not expressed or were expressed at a very low value in this sample type. Identification of miRNA precursors was performed in a previous study by mapping the mature miRNA sequences to the *Pinus* PGI_v9.0_032811 unigene sequences [38]. Predicted precursor sequences were found for 11 DE miRNAs, including 4 miRNAs with 2 precursors.

### 3.3. Gene Ontology Annotation of Target Genes of Differentially Expressed microRNAs

There were 109 target genes for the 15 down-regulated miRNAs and 396 target genes for the 43 up-regulated miRNAs) identified. GO annotation of the predicted target genes of up-regulated miRNAs indicated that the most common functions in the biological process domain were related to different cellular and metabolic processes, and response to stimulus, e.g., response to stress, to chemicals, cellular response to stimulus, response to abiotic stimulus, biotic stimulus and anatomic structure development, regulation of molecular function, small molecule metabolic process, development process involved in reproduction and others (Appendix A). The most common GO annotations in the molecular function domain were catalytic activity (oxidoreductase activity, transferase activity, electron transfer activity, transcription regulator activity, hydrolase activity, and electron transfer activity), binding, including small molecule binding, ion binding, nucleotide, DNA, RNA, mRNA, and molecular function regulation (Appendix A).

GO annotation of the predicted target genes of down-regulated miRNAs indicated that the most common functions in the biological process domain (Appendix A) were also related to response to stimulus (cellular response to stimulus, response to stress, response to biotic stimulus, response to external stimulus, response to chemical, response to other organisms, and others), cellular and metabolic process, regulation of cellular, metabolic, catabolic process, developmental process, cell division, gene silencing, etc. The most common GO annotations in the molecular function domain were binding (ion, small molecule binding, protein binding, DNA, RNA, lipid binding), catalytic activity, transporter activity, translation regulator activity, and antioxidant activity (superoxidase dismutase activity) (Appendix A).

GO annotations of the most highly represented target genes encoding enzymes included glycosylases, hydrolases, oxidoreductases, laccase, transferring phosphorus-containing groups, and transferases, and were the most represented for DE down-regulated miRNAs after MeJA treatment. The 30 most represented GO categories for target genes of DE down-regulated miRNAs after MeJA treatment are shown in Figure 2.

The 30 most represented GO categories for target genes of up-regulated miRNAs after MeJA treatment are shown in Figure 3. Some of these categories were also represented in the target genes of down-regulated miRNAs, but some categories such as acyltransferases, sucrose synthase, and lyases, were present only in the target genes of down-regulated miRNAs.

GO term enrichment indicated that target genes of up-regulated miRNAs were enriched for defense response and response to stimulus GO categories, as compared to target genes of down-regulated miRNAs. Target genes in the GO categories of lignin metabolic and biosynthetic process, protein quality control for misfolded or incomplete folding, and molybdate ion transport were only targeted by down-regulated miRNAs (Figure 4). In contrast, target genes in the GO categories electron transport chain, cation transport, and ion membrane chain were only targeted by up-regulated miRNAs.

More target genes with molecular functions for binding, such as nucleotide and ribonucleotide binding, small molecule binding, and purine nucleotide binding were found as targets of down-regulated miRNAs compared to up-regulated miRNAs (Figure 5.). Genes with molecular functions such as oxidoreductase activity and hydroquinone: oxygen oxidoreductase activity were not found as targets of up-regulated miRNAs (Figure 5.). The opposite situation was observed for genes that are involved in electron transfer activity, which were targets only for up-regulated miRNAs.

### 3.4. Target Genes of Differentially Expressed Conifer-Specific microRNAs

A total of 315 target genes were found for the 41 DE conifer-specific miRNAs from 17 families (Appendix A). For each DE miRNA between 4 (miR1314) to 43 target genes (miR1313) were identified.

### 3.5. Identification and Expression Analysis of Target Genes of Differentially Expressed microRNAs from Transcriptome Data

Analysis of the transcriptome revealed 389 gene transcripts containing 562 target sites targeted by 57 of the 58 DE miRNAs belonging to 29 miRNA families (Appendix A). A total of 250 target genes with 138 different GO term annotations were found for the 41 DE conifer-specific miRNAs from 17 families (Figure 6). The other DE miRNAs (miR159, miR164, miR169, miR396, miR397, miR398, miR408, and miR535) were considered as conserved miRNAs, which are also found in angiosperm species (as reported in miRBase).

A network of DE conifer-specific miRNAs and target genes identified from transcriptome sequencing was constructed. For each DE conifer-specific miRNA, one or more target genes were identified, resulting in a total of 273 target sites (Figure 7). The lowest numbers of target genes were found for miR1314, miR3701, and miR11490, while the highest numbers were found for miR1313, miR946, and miR11487 with 43, 36, and 31 target genes, respectively. Target genes with unknown functions were not represented in the network analysis.

### 3.6. Resistance-Related Target Genes of Differentially Expressed microRNAs from Transcriptome Data

Transcriptome analysis identified 68 disease resistance, resistance, tobacco mosaic virus (TMV) resistance protein, Toll/Interleukin-1-Nucleotide-binding site–leucine-rich repeat (TIR-NBS-LRR), LRR receptor-like serine/threonine-protein kinase, putative CC-NBS-LRR protein, and putative NBS-LRR protein target transcripts (referred to further in the text as disease-resistance transcripts). These disease-resistance transcripts were annotated with 29 target gene GO term descriptions (Appendix A) and were targeted by 26 DE miRNAs from 14 miRNA families (miR164, miR946, miR947, miR950, miR1312, miR1313, miR3693, miR6171, miR11504, miR11511, miR11532, miR11544, miR11466, and miR11551). Of these 14 miRNA families, 12 families (24 miRNAs) are conifer-specific (except for families miR6171 and miR164).

### 3.7. Analysis of Differentially Expressed microRNAs Targeting Differentially Expressed Gene Transcripts

DE transcriptome analysis revealed 44 gene transcripts as DE target genes (with 77 target sites) of 38 DE miRNAs from 24 families (Appendix A). Of these DE miRNAs and DE target genes, nine up-regulated miRNA sequences from seven families (miR159, miR164, miR169, miR1313, miR00002, miR00028, and miR11504) targeted seven down-regulated target genes. Eight down-regulated miRNA sequences from five families (miR397, miR535, miR946, miR950, and miR11460) targeted nine up-regulated target genes (Appendix A). For 26 up-regulated miRNAs from 17 families (miR159, miR164, miR396, miR408, miR946, miR947, miR1312, miR1313, miR3693, miR6171, miR11452, miR11487, miR11511, miR11544, miR00002, miR00024, and miR00028) and three down-regulated miRNAs from three families (miR535, miR11466, and miR11532), target gene expression changes were in the same direction. From these, one member from each of the families miR164, miR1313, miR00002, and miR00028 and two members from miR159 were found in both the up-regulated miRNAs and down-regulated target gene transcript group as well as the up-regulated miRNA and up-regulated target gene transcript group. One member from each of the families miR535 and miR11466 was found in the down-regulated miRNA and up-regulated target gene transcript group, as well as the down-regulated miRNA and down-regulated target gene transcript group. All DE miRNAs and DE target gene interactions in opposite miRNA-target gene transcript directions are presented in Figure 8.

## 4. Discussion

MiRNAs have a significant function in the transcriptional and post-transcriptional regulation of genes that are involved in the regulation of plant development, cell communication, signaling, and plant defenses. MeJA is involved in plant defense signaling and can stimulate induced resistance, a defense mechanism in plants against various biotic and abiotic factors. MeJA can be used to induce defense responses prior to pest exposure and therefore can play a role in plant protection strategies. Previous studies concluded that MeJA can modify terpene composition in Norway spruce and decrease damages caused by the insect pest *Hylobius abietis* L. and the fungal pathogen *Endoconidiophora polonica* (Siemaszko) Z.W.de Beer, T.A.Duong & M.J.Wingf., but the extent of protection can vary between different Norway spruce clones [52]. Other studies of Norway spruce [31] and Chinese yew [30] have reported that MeJA treatment profoundly alters the expression of miRNAs as well as the expression of miRNAs in response to a subsequent wounding challenge.

Our analyses revealed that MeJA treatment altered the expression of 17 DE miRNAs and their 15 target genes (a total of 21 DE miRNA-target gene combinations) in opposite directions, as expected, but we did not find the expected relationship between changes in the expression of 29 miRNAs and of their predicted 31 mRNA targets (a total of 56 miRNA-target gene combinations). Possibly, the expected gene expression changes may have been observed more frequently if the samples for miRNA analysis and transcriptome analysis were collected at different time points after treatment with MeJA. These unexpected expression patterns could also be explained by the fact that miRNAs can affect the target gene regulation not only by cleavage, but also by binding to and inhibiting the translation of target mRNA, and therefore the targeted transcripts are still observed in the transcriptome, but the genes are not translated to proteins.

The most highly expressed miRNA families identified in this study were conifer-specific miRNAs. These results were different from a previous study in conifers, MeJA-treated cultured *Taxus chinensis* (Pilg.) Rehd. cells, where the most expressed miRNAs were the highly conserved families miR156, miR172, and miR168 [30]. This could be explained by the fact that there are relatively few miRNA expression studies in conifers, and miRNA sequences available in databases have mostly been identified in other, non-coniferous species. Prior to 2009, miRNAs unique to gymnosperms were only reported in *Pinus contorta* Douglas ex Loudon [46], therefore a previous study identified the families miR1310 and miR1314 as conifer-specific miRNAs because the identification of *T. chinensis*-specific miRNAs was complicated due to the lack of a *T. chinensis* genomic sequence and the paucity of conifer miRNA sequences in databases [30]. Increased use of high-throughput sequencing strategies and knowledge about the expression of miRNAs in a range of conifer species and conditions have allowed the identification of previously unidentified conifer-specific miRNAs.

### 4.1. Differentially Expressed microRNAs in Conifers After Methyl Jasmonate Treatment

DE up- and down-regulated miRNAs were not only observed between miRNA families but also among different members of the same family, as also previously reported in other studies in conifers, which indicates potential functional divergence within miRNA families. For example, a total of 654 members were identified for the conifer-specific miRNA family miR950, but some miR950 family miRNAs were highly expressed in both MeJA-treated and control samples (e.g., *n* = 26,002, *n* = 24,712, *n* = 1203, etc.), while other miR950 members had low expression levels (e.g., *n* = 85, *n* = 50, *n* = 1 etc.). Similar results for different miRNA families were also reported in Chinese yew [30].

Of the 58 DE up- and down-regulated miRNAs identified between MeJA-treated and control samples, 41 were conifer-specific miRNA families, and 17 were conserved miRNAs, which are highly conserved among plants [51]. These DE miRNAs included the miR396 family, which has been found in all vascular plants (tracheophytes), while miR164, miR169, miR397, and miR398 are present in all seed plants (spermatophytes), and miRNA families miR159 and miR408 previously were identified in all green plants, bryophytes, as well as in *P. sylvestris*, *Agave angustifolia* Haw., and *B. pendula* [38,51,53,54]. MiR535 has been found in *P. sylvestris*, *Physcomitrella patens* Hedw., *P. abies*, and in angiosperm species such as *Vitis vinifera* L., *Carica papaya* L., *Salix bicolor* Ehrh. ex Willd., and *Oryza sativa* L. The miRNA families miR1312, miR1313, miR1314, miR3701, miR946, miR947, and miR950 have previously been described in at least three conifer species, including *P. sylvestris*. The DE miRNAs miR3693, miR11466, miR11452, miR11487, miR11490, miR11504, miR11511, miR11532, miR11544, and miR11551 were previously identified only in Pinaceae (*Picea abies*) [31].

Comparing our data with other studies of miRNA expression in conifers after treatment with MeJA, of the 29 DE miRNA families from *P. sylvestris*, 9 families were identified in *T. chinensis* [30], 6 families in Taxus × media [55], and 28 families in *P. abies* [31] were also DE. However, the expression of miR164 in both *Taxus* studies was in the opposite direction than in our results in *P. sylvestris.* In addition, the expression of seven DE *P. abies* miRNAs (miR398, miR11487, miR11511, miR169, miR950, miR169, and miR3693) was also in the opposite direction than in *P. sylvestris* (Table 2). From the miRNA families identified to be DE in two of more conifer studies, only miR169, miR398, miR950, miR3693, miR11487, and miR11551 were the same direction, but in the case of miR396, expression was in a different direction only in *P. sylvestris*. These differences could be explained by species-specific miRNA expression patterns, as well as by differences in the age or developmental stage of the analyzed plant material, tissue type, different MeJA concentrations during treatment and sample collecting time points, as well as due to the expression of different family members. Data analysis parameters could also affect the identification of DE miRNAs.

### 4.2. Predicted Target Genes for Differentially Expressed Down-Regulated microRNAs

The prediction of target genes revealed a large number of gene families with diverse biological and molecular functions. The predicted target genes of DE down-regulated miRNAs included laccases, TIR/NBS/LRR and other disease-resistance proteins or homologs, heat stress transcription factor B-4b or protein, tau class glutathione S-transferase, serine carboxypeptidase-like 50, zinc finger A20, AN1 domain-containing stress-associated protein 8, and others, which are well known to participate in pathogen sensing and plant defenses.

Laccases are a family of copper-containing oxidoreductases that play general roles in plant growth and development. Expression levels of laccases can be regulated by transcription factors (myeloblastosis—MYBs and Basic leucine zipper—bZIPs) and miRNAs (miR397, miR408, and miR857) [56]. Plant laccases can contribute to plant biotic and abiotic stress resistance [56,57]. Recent studies have shown that the Norway spruce laccase PaLAC5 is likely to be involved in induced defense responses. Expression analyses showed that expression of PaLAC5 responds specifically and strongly in close proximity to the site of *Heterobasidion parviporum* Niemelä & Korhonen inoculation [58]. Our study identified one down-regulated miRNA (miR397) that targets 17 laccase genes while the up-regulated miR408 found in our study targeted other genes, unrelated to laccases, such as plantacyanin, Basic blue protein, Uclacyanin I, Chemocyanin precursor, and 60S acidic ribosomal protein.

Down-regulated DE miRNA-target genes included several disease-resistance proteins (putative TIR-NBS-LRR protein, probable disease-resistance protein At1g61300 isoform X1, TMV resistance protein N-like, disease-resistance protein RFL1-like, disease-resistance protein At4g27190-like, disease-resistance protein TAO1-like, TMV resistance protein N-like, etc.), heat shock 70 kDa protein 15-like, zinc finger protein GIS2-like and zinc finger CCCH domain-containing protein 27 (reported to respond to oxidative stress), 7-deoxyloganetin glucosyltransferase-like, BI1-like protein, transport protein genes (MOS14 isoform X1, NPF, molybdate anion), etc.

TIR-NBS-LRR and other NBS-LRR genes are involved in defense responses to pathogen infection and disease resistance. The NBS-LRR gene family may confer resistance to biotic aggressors including viruses, bacteria, fungi, nematodes, insects, and abiotic stresses such as drought [59]. Target disease-resistance protein genes have been identified in this study for DE up- and down-regulated miRNAs from the transcriptome data, although a total of 46 disease target resistance protein genes were identified from the transcriptome data, including TIR-NBS-LRRs, targeted by 20 DE miRNAs from 13 families. A study in Norway spruce [60] reported that 24 miRNA families (miR482, miR946, miR950, miR951, miR1311, miR1312, miR2118, miR3697, miR3701, miR3709, miR3710, miR11425, miR11466, miR11476, miR11482, miR11506, miR11511, miR11519, miR11523, miR11528, miR11532, miR11540, miR11546, and miR11551) are involved in the targeting of LRR target genes. Of these, seven DE miRNA families (miR946, miR950, miR1312, miR11466, miR11511, miR11532, and miR11551) were also found to be DE in our data and were associated with disease-resistance protein genes. The families miR164, miR947, miR1313, miR1504, miR3693, miR6171, and miR11544, were found to be associated with disease-resistance protein genes in our data but were found in the Norway spruce study as being involved in targeting of LRR target genes.

The down-regulated miR11466 and miR946 families were identified as targeting TIR/NBS or NBS-LRR genes in our data. Both of these miRNA families target either the linking NBS region (miR946) or the region between the NBS and LRR domains (miR11466) in Norway spruce [60]. In a study on *Larix olgensis* A. Henry, TIR/NBS-LRR target genes were also reported to be targeted by some members of the miR950 family [61], similarly to the results from this study.

Our results identified three down-regulated miRNAs from the miR11532 and miR397 families, which targeted eight heat shock transcriptional factors or proteins. Heat shock transcription factors are present in the majority of plant species and play central roles in thermotolerance, transgenerational thermomemory, and many other stress responses [62].

Glutathione *S*-transferases are involved in primary and secondary metabolism, signaling and stress metabolism, herbicide detoxification, plant protection against ozone damage, heavy metals, xenobiotics, microbe infections, various aspects of seedling development, and others [63]. Only one miR535 family member with one target gene was associated with glutathione *S*-transferase in our study. Previous studies have not reported an association between miR535 and glutathione *S*-transferases.

The serine carboxypeptidase-like protein family plays a major role in stress response, growth, development, and pathogen defense [64]. One member of the miR11490 family found to be DE in this study targeted a serine carboxypeptidase-like protein target gene. However, as for the glutathione *S*-transferase genes, previous studies have not reported an association between miR11490 and serine carboxypeptidase-like protein genes.

Zinc finger domains present in the proteomes of many different organisms enable different proteins to interact with or bind DNA, RNA, or other proteins. Proteins containing zinc finger domains have an important role in eukaryotic cells, regulating different signal transduction pathways and controlling processes such as development and programmed cell death [65]. We found four zinc finger domain protein target genes targeted by two members from each of the miR397 and miR11466 families. A study on *L. olgensis* [61] reported that the miR166 family also targeted zinc finger domain (class III HD zip) protein target genes, but the miR166 family was not DE in our study.

Transport processes in plants are significant for numerous cellular processes: nutrient acquisition, development, cellular homeostasis, communication, and coordination of responses [66]. A previous report described ABC transporter genes in a study on *L. olgensis* as associated with the miR164 family [61], but in Norway spruce [60] several different transporter genes were associated with miR11408, miR11451, miR11517, miR11531, miR11532, and miR11564, from which only miR11532 was found to be DE in this study. However, there are 11 miRNAs from seven families in our data (miR164, miR169, miR946, miR950, miR1313, miR00028, and miR11532) that target genes involved in transport functions.

### 4.3. Predicted Target Genes for Differentially Expressed Up-Regulated microRNAs

Predicted target genes for DE up-regulated miRNAs included genes encoding lipase, dehydrin, resistance gene homolog or disease resistance-associated proteins, TIR, TIR/NBS, probable TIR/P-loop/LRR, CC-NBS-LRR resistance-like protein, LRR protein, LEA protein, transcriptional regulator proteins (GAMYB-like, GHMYB10) as well as genes encoding basic blue copper protein, thaumatin-like protein, Sucrose synthase, AINTEGUMENTA-like protein, anthocyanidin reductase, and others.

Among the DE up-regulated miRNA-target genes, a large proportion were various kinases. Protein kinases are enzymes, and in plant genomes, are among the largest gene families. They regulate the activity of their target proteins by catalyzing the reversible phosphorylation of the amino acids serine, threonine, and tyrosine [67].

A total of 33 DE-regulated target genes from the transcriptome data in our study were annotated as kinases and were targeted by 18 DE up-regulated miRNAs from 11 families (miR159, miR164, miR408, miR947, miR1312, miR1313, miR6171, miR00028, miR11487, miR11544, and miR11551). These target genes also include an LRR receptor-like serine/threonine-protein kinase targeted by three DE members from the miR947 family. LRR receptor-like serine/threonine-protein kinases are involved in many plant responses such as development, growth, hormone perception, and the response to pathogens [68]. However, other miRNAs associated with LRR receptor-like serine/threonine-protein kinase protein genes have been identified in Norway spruce, such as miR11409, miR11421, miR11422, miR11432, miR11433, miR11434, miR11435, miR11436, miR11437, miR11440, miR11447, miR11448, miR11549, and miR15551 [60]. These were not found to be DE in this study, and the DE miR947 miRNAs were not identified in the Norway spruce study [60].

Lipases play specific functions in plant-microbe interactions and defense responses and are activated in plant cells by pathogen infection. Lipases cleave or modify lipid substrates in a number of subcellular structures [69]. This study identified five up-regulated miRNAs from two families, miR947 and miR11504, that targeted two lipase target genes from predicted target genes and one miR1312 family member targeted a lipase gene identified from the sequenced transcriptome. Previously, miR947 has been associated with the lipase target gene in male gametophytes of loblolly pine [70].

Dehydrins have a major role in plant responses to a range of stresses that cause dehydration, including drought, freezing, salinity, and others. The dehydrin gene family is very large in conifers, indicating the importance of this family for physiological functions related to stress tolerance [71]. Analysis of Scots pine miRNA putative dehydrin target genes identified five predicted target genes targeted by one miRNA from the miR11551 family.

AINTEGUMENTA-like proteins are members of the APETALA 2/ETHYLENE RESPONSE FACTOR (AP2/ERF) domain family, and are transcription factors regulating plant growth, development, and responses to abiotic stresses [72]. Our analysis identified one predicted AINTEGUMENTA-like gene targeted by one miRNA family (miR947) with three isomiRs. Some miR156 and miR172 isoforms are involved in the regulation of phase transition in a number of species [73]. MiR156 suppresses the expression of miR172 via interactions with the SPL9 and SPL10 genes, resulting in increased expression of miR172-targeted AP2-like transcription factor family genes such as APETALA2 (AP2) and others. The miR947 family has not been previously reported to target AP2 genes. The miR947 family, which was previously reported to be conifer-specific [74,75], was also found in *Betula pendula*, but with low expression, and was not found in juvenile tissue samples [50].

Thaumatin-like (TLP) proteins are pathogenesis-related proteins involved in responses to pathogen infection. Some plant TLPs exhibit strong antifungal activity while others have no or only weak antifungal activities [76]. Analysis of up-regulated miRNA putative target genes identified six TLP genes targeted by only one miR11487 family with four miRNAs. In a previous study of Norway spruce, predicted miR11487 mRNA targets were described as containing the epigenetic modification reader domain PWWP or the Putative S-adenosyl-L-methionine-dependent methyltransferase domain, which are characteristic of DNA methyltransferases [31], but the miR11487 family has not been previously associated with TLP target genes.

Basic blue copper proteins are small and soluble copper-containing proteins that facilitate monoelectron transfer processes and whose role is to shuttle electrons from an electron donor to an electron acceptor in bacteria and plants [77,78]. In this study, six basic blue copper target genes were predicted to be targeted by one member of the miR408 family. In a previous study on common beans, a basic blue copper protein gene was also reported to be targeted by the miR408 family [79].

Sucrose synthase plays a general role in sugar metabolism, catalyzing the reversible cleavage of sucrose into the subunits of fructose and either uridine diphosphate glucose or adenosine diphosphate glucose [80]. There is no previously published information about miRNAs associated with sucrose synthase target genes, but our results indicated that seven sucrose synthase genes were targeted by one member of the conifer-specific miR1313 family and one miR1313 member for one target gene.

The late embryogenesis abundant (LEA) protein family plays a major role in enhancing freezing tolerance. The identification, molecular functions, and regulatory networks of the LEA protein family have not been extensively studied in conifers or gymnosperms [81]. The LEA proteins stabilize cell membranes and prevent cell dehydration and crystallization under various stresses [82,83]. Overexpression of LEA genes in many plants can improve their tolerance to abiotic stresses [81]. In this study, miR950, miR1312, and miR1313 families were predicted to target LEA predicted genes. A total of four LEA target genes were predicted to be targeted by two conifer-specific members from the miR1312 family and one member from each of the families miR1313 and miR950.

Transcription factors, which possess DNA-binding domains, play a significant role in controlling the transcription regulation and developmental processes, as well as responses to environmental cues in plants [84]. A total of nine transcription factor target genes from the transcriptome data were found to be targeted by eight miRNAs from four families (miR159, miR00028, and miR11487). Of these, miR159 and miR11487 targeted genes associated with regulation of the mediator protein complex, while miR11531 is associated with the calmodulin-binding transcription activator 5 isoform X2, but miR00028 with transcription factor 3. Mediator is a large protein complex that binds RNA polymerase II and controls transcription from class II genes. It binds RNA polymerase II and is probably critical for basal transcription, but it also mediates activation and repression of transcription [85]. None of these miRNAs were found associated with transcription factors in a study of miRNA expression in Norway spruce [60].

MYB genes are transcription factors, that have a wide range of biological functions and participate in many processes in plants, including regulating plant growth and development, cell morphogenesis, regulating primary and secondary metabolic reactions, and responding to abiotic and biotic stresses [86]. In *Arabidopsis thaliana* L., miR159 regulates the timing of the vegetative phase change by suppressing the expression of MYB33, preventing it from hyperactivating miR156 expression during the seedling stage [87]. Recent studies in *T. chinensis* have demonstrated that miR159 is one of the most abundantly expressed miRNAs during different developmental periods and it effectively represses GAMYB and thus regulates the growth and development of *T. chinensis* [88]. In this study, one GAMYB target gene targeted by three members from the miR159 family and one GHMYB target gene targeted by two members of the miR11504 family were identified.

### 4.4. Differentially Expressed Pinus sylvestris Genes Targeted by Differentially Expressed microRNAs

GO annotation found that the molecular function descriptions of DE target genes included homeotic protein female sterile-like isoform X1 (required maternally for proper expression of other homeotic genes involved in pattern formation), glucan endo-1,3-β-glucosidase-like, Cytochrome P450 CYPC, 3-oxo-Delta (4,5)-steroid 5-β-reductase-like, putative clathrin assembly protein At2g25430, anthocyanidin reductase, UDP-glycosyltransferase 74E2-like and others. Almost all DE transcriptome target genes were unique and targeted by a specific miRNA. Disease-resistance or LRR target genes were only targeted by miR947. It has been previously reported that miR947 is associated with a probable role in signaling during pathogen recognition, and the subsequent activation of plant defense mechanisms [89].

The β-1,3-glucanases are widely distributed in higher plants and play an important role in plant defenses against fungal pathogens and plant physiological and developmental processes [90]. The β-1,3-glucanase gene in wheat is activated by infection by the pathogen *Puccinia striiformis* f. sp. Tritici Wes., as well as application of salicylic acid (SA), MeJA, and ethylene [91]. From our transcriptome data, a glucan endo-1,3-β-glucosidase-like gene was targeted by four miRNAs (two from each of the miR1313 and miR1532 families).

Cytochrome P450 plays an important role in many cell processes that affect plant growth and development as well as plant defense. These interactions are mediated via phytoalexin biosynthesis, hormone metabolism, and the synthesis of other secondary metabolites. Phytoalexins are antimicrobial compounds that are synthesized in plants for protection against plant pathogens [92,93]. Cytochrome P450s (CYPs) are also involved in the regulation of JA and MeJA signaling pathways for plant defense. Ref. [25] found that CYP74A (allene oxide synthase) catalyzes the dehydration of hydroperoxide to an unstable allene oxide, and is crucial for activating wound-induced defense against biotic attacks. [94,95] clarified that the CYP94 family is involved in jasmonoyl-isoleucine turnover, which is mediated by the ω-oxidation pathway, activating immune function in plant defense responses. CYPs are also important for the synthesis of compounds responding to biotic stresses, such as terpenoids, alkaloids and cyanogenic glucosides. In Sitka spruce, CYP720B4 is involved in defense responses to insect attack, via the diterpene metabolism pathway, and catalyzes multiple oxidation steps in resin acid biosynthesis [96].

The regulation of P450 proteins as a response to a range of plant stresses promotes the establishment of complex signaling webs. and they are potential candidates for increasing stress tolerance in plants. In our transcriptome data, a cytochrome P450 CYPC gene was targeted only by one miRNA member from the miR164 family.

## 5. Conclusions

MeJA is a plant signaling molecule that broadly affects gene expression, impacting plant growth and development, response to pathogen attack, wounding, and plant responses to abiotic and biotic stresses. Previous studies have shown that treatment with MeJA up and down-regulates miRNA and gene expression in different plant and tree species.

In our study, a large number of miRNAs and target genes were identified from miRNA and transcriptome sequencing from Scots pine needles after treatment with MeJA. 43 miRNAs were up-regulated and 15 were down-regulated with statistical significance (*p*-value ≤ 0.05) and fold change > 1.5. Most of the DE miRNAs (41 from 17 families) were conifer-specific miRNAs. Potential miRNA-target genes were found by searching for complementary regions between DE miRNAs and the DFCI pine gene index as well as the transcriptome sequenced in this study. These target genes included 68 disease-resistance genes, TMV resistance protein, TIR-NBS-LRR, LRR receptor-like serine/threonine-protein kinase, putative CC-NBS-LRR protein, and putative NBS-LRR protein, which were targeted by 14 DE miRNA families, from which almost all (12 families) are conifer specific.

Our study indicated that MeJA treatment altered the expression of 17 miRNAs and their 15 target genes (a total of 21 DE miRNA-target gene combinations) in opposite directions, as expected, but we did not find the expected relationship between changes in the expression of 29 miRNAs and of their predicted 31 mRNA targets (a total of 56 miRNA-target gene combinations).

GO annotation found that most of the molecular function descriptions of DE target genes identified from the transcriptome, which were expressed in the opposite direction to their corresponding miRNAs, included homeotic protein female sterile-like isoform X1, glucan endo-1,3-β-glucosidase-like, Cytochrome P450 CYPC, 3-oxo-Delta (4,5)-steroid 5-β-reductase-like, putative clathrin assembly protein At2g25430, anthocyanidin reductase, UDP-glycosyltransferase 74E2-like, and others.

Some of the genes targeted by conifer-specific miRNAs have been previously reported to be targeted by other miRNAs in angiosperms, indicating that the miRNA-target gene regulation system can vary between species. This is also supported by the fact that only some miRNAs from one family were differentially expressed. This provides for flexible regulation of gene expression, where one miRNA can target multiple genes and vice-versa. In addition, while the majority of miRNA sequences currently found in databases are from angiosperms, additional studies may reveal that the conifer-specific miRNAs identified in this study and other studies are also found in other plant clades. Unfortunately, successful miRNA-target gene analysis is also hampered by the fact that many conifer genes are not annotated. Undoubtedly, the study of additional conifer-specific target genes would provide greater knowledge and insight into miRNA-target gene regulation in conifers.

## Figures and Tables

**Figure 1 genes-16-00026-f001:**
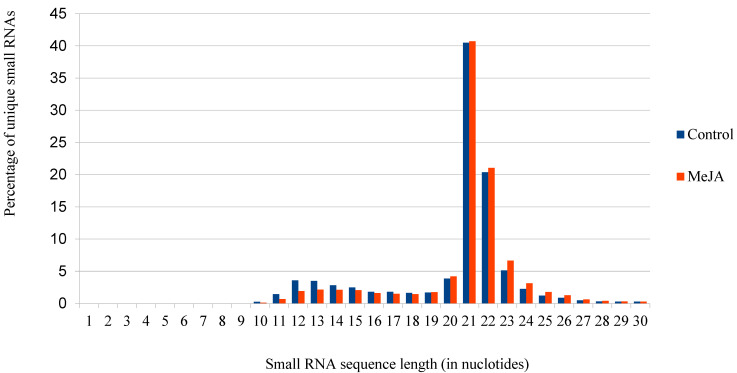
Length distribution of small RNAs in the control and methyl jasmonate-treated samples.

**Figure 2 genes-16-00026-f002:**
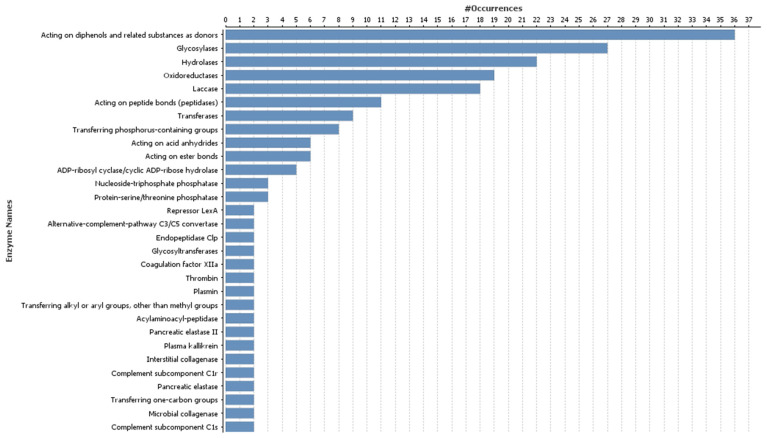
The 30 most highly represented enzyme Gene Ontology categories for target genes of differentially expressed down-regulated microRNAs.

**Figure 3 genes-16-00026-f003:**
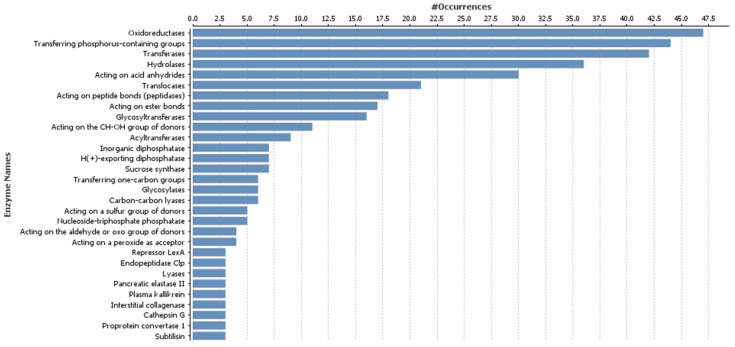
The 30 most highly represented enzyme Gene Ontology categories for target genes of differentially expressed up-regulated microRNAs.

**Figure 4 genes-16-00026-f004:**
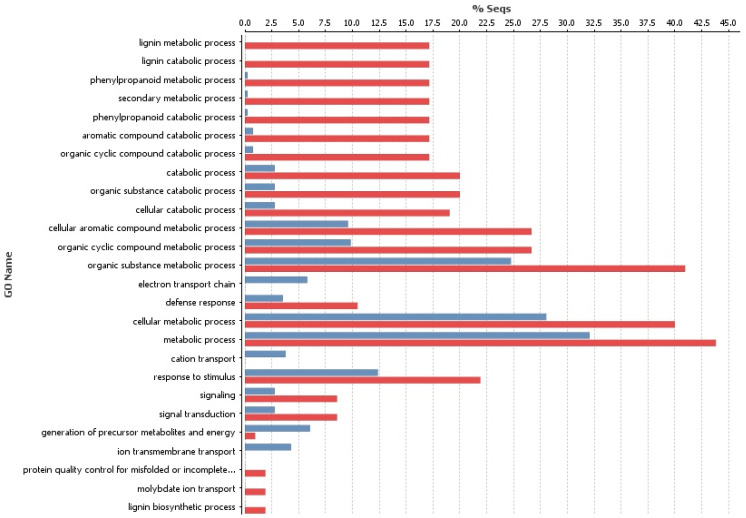
Gene Ontology term enrichment for biological processes for target genes (targets from psRNATarget tool) for up (blue bars) and down-regulated (red bars) miRNAs (control vs. methyl jasmonate-treated).

**Figure 5 genes-16-00026-f005:**
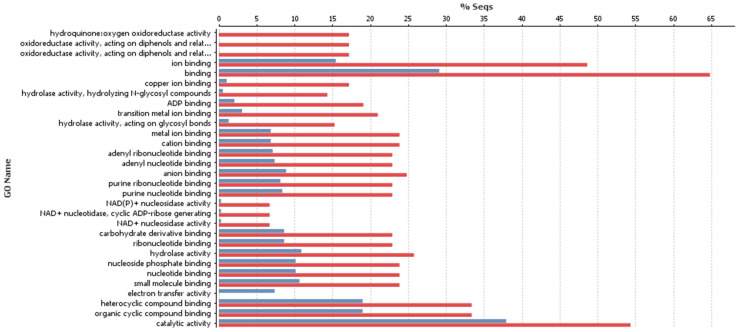
Gene Ontology term enrichment for molecular functions for target genes (targets from psRNATarget tool) for up- (blue bars) and down-regulated (red bars) microRNAs (control vs. methyl jasmonate-treated).

**Figure 6 genes-16-00026-f006:**
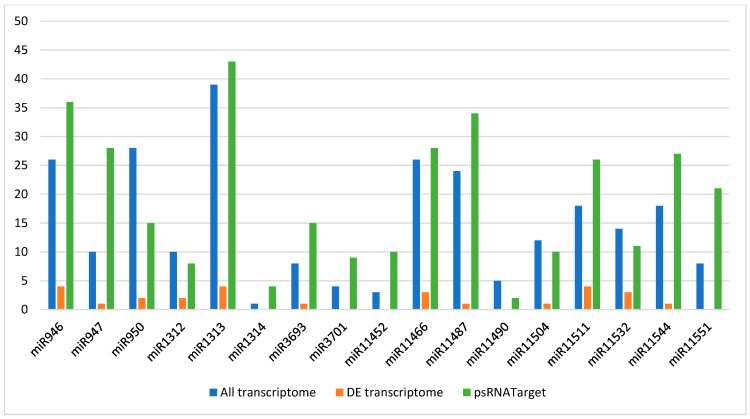
Number of genes targeted by conifer-specific differentially expressed microRNAs. Blue bars represented target genes for differentially expressed conifer-specific microRNAs from all transcriptome data; orange bars—target genes for differentially expressed conifer-specific microRNAs from differentially expressed transcriptome data; green bars—target genes for differentially expressed conifer-specific microRNAs from *Pinus* transcript sequence input of the psRNATarget-Plant Small RNA Target Analysis Server.

**Figure 7 genes-16-00026-f007:**
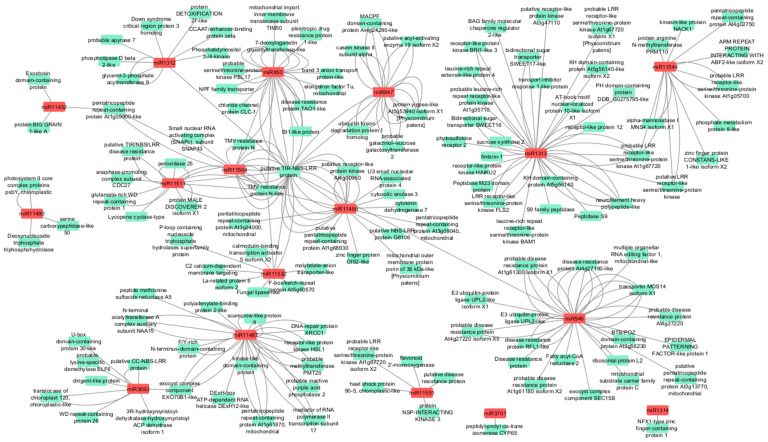
A network of differentially expressed conifer-specific microRNAs and their target genes identified in transcriptome sequencing. Grey lines represent target gene and microRNA interactions to each target gene’s Gene Ontology Term description; red nodes—conifer-specific differentially expressed microRNAs; green nodes—Gene Ontology Term descriptions of target genes from the entire transcriptome. More than one connecting line indicates multiple target genes or microRNA isomiRs.

**Figure 8 genes-16-00026-f008:**
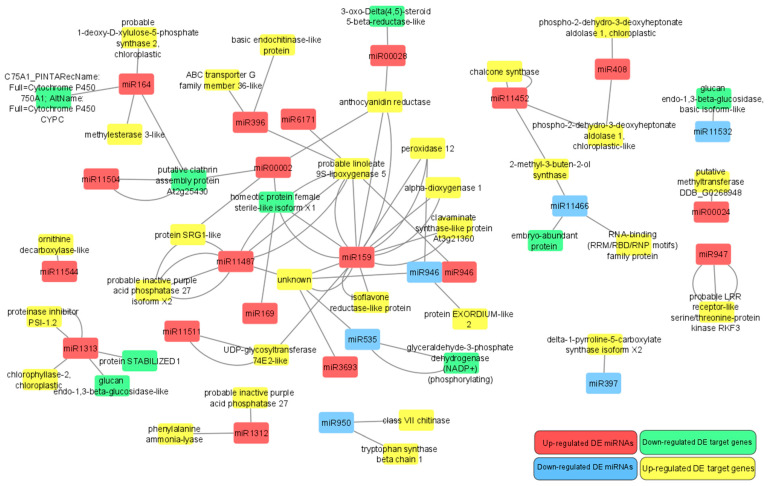
Network of differentially expressed microRNA and differentially expressed target gene interaction. Grey lines represent target gene and microRNA interactions to each target gene’s Gene Ontology Term description, red nodes—up-regulated miRNAs, blue nodes—down-regulated miRNAs, green nodes—down-regulated target genes, yellow nodes—up-regulated target genes. More than one connecting line indicates multiple target genes or microRNA isomiRs.

**Table 1 genes-16-00026-t001:** Statistically significantly differentially expressed up- and down-regulated microRNAs identified in methyl jasmonate-treated samples compared to control samples.

microRNA Family	microRNA Sequence	Mature microRNA Length (in Nucleotides)	Count of Precursor-microRNAs	Precursor-microRNA Accession Number
Up-regulated				
miR00002	GTTCGGAGAAATGCGTGCTCG	21	1	TC188772
miR00024	CGTGTTCCCAGGTCGCCCGG	20	1	TC188772
miR00024	CGTGTTCCCAGGTCACCCAG	20	1	TC188772
miR00024	CGTGTTCCCAGGTCGCCCCGG	21	1	TC188772
miR00028	ATGTGAGAAAAGGGTTTGTGG	21	1	AW010185
miR11452 *	TGGGAGCGATCGATGAGGTGTT	22	-	-
miR11466 *	CGATCTCCAGAAGACACTTGT	21	-	-
miR11487 *	CATATTGATCGCCTTCTCAGT	21	-	-
miR11487 *	TCATATTGATCGCCTTCTCAG	21	-	-
miR11487 *	TGAGAAAGCGATCAACATGAC	21	-	-
miR11487 *	TGAGAAGGCGATCAACATGAC	21	-	-
miR11487 *	TGAGAAGGCGATCAACATGAT	21	-	-
miR11487 *	TGAGAAGGCGATCAACATGC	20	-	-
miR11504 *	AATGAGCTCCTCCTCATGTCC	21	-	-
miR11504 *	AATGAGCTCCTCCTCATGTCT	21	-	-
miR11511 *	TCCAACGAAGATCAGAAAGG	21	-	-
miR11511 *	TCCAACGAAGATCAGAAGGC	20	-	-
miR11511 *	TCCAACGAAGATCAGAAGGCGT	22	-	-
miR11511 *	TCCAACGAAGATCAGAAGGCTC	22	-	-
miR11511 *	TCCACCGAAGATCAGAAGGTTC	22	-	-
miR11511 *	TCCACCGAAGATCAGAAGGTTTT	23	-	-
miR11544 *	TGGAGCTGTTGTCACTCCACT	21	-	-
miR11551 *	TGTTTTGCTTTCCCTCCGCAAT	22	-	-
miR1312 *	TCGGAGAGAATATGGCGAGAT	21	-	-
miR1313 *	AACAATAATTTCAGTGGAAGA	21	2	CO167297, DR013032
miR1314 *	TCGGCCTCGAAATGTTAGGAGAA	23	-	-
miR159	TTTGGTTTGAAGGGAGCTCCA	21	-	-
miR159	CTTGGATTGAAGGGAGCTCC	20	1	DR078315
miR159	TTTGGTTTGAAGGGAGCTCT	20	-	-
miR164	TGGAGAAGCAGGGCACGTGCG	21	-	-
miR169	GGCAAGTTGTTCTTGGCAAAG	21	-	-
miR3693 *	CTGAACTGCTTATAGATGGGA	21	-	-
miR3693 *	TGAACTGCTTATAGATGGGAG	21	-	-
miR396	CTCAAAGAAAGCTGTGGGAAA	21	-	-
miR398	ACGTGTTCCCAGGTCACCCCA	21	2	TC184482, TC188772
miR398	CGTGTTCCCAGGTCACCCCAG	21	2	TC184482, TC188772
miR408	TGCACTGCCTCTTCCCTGGCT	21	2	TC159922, TC180357
miR6171	TGTGGATTGCTGAAGGATTTA	21	-	-
miR946 *	TGTGGATAGAGAAGGGTTAGT	21	-	-
miR947 *	CATCGGAATCTGTTACTGTTTT	22	-	-
miR947 *	CATCGGAATCTGTTACTGTTTC	22	-	-
miR947 *	CATCGGAATCTGTTACTGTTT	21	-	-
miR950 *	TCTGGTCCTCGGTGGTTTATGAAT	24	-	-
Down-regulated				
miR11466 *	CACTTCCAATAGACACTTGTT	21	-	-
miR11466 *	TCTACTTCCACAAGACACTTGCC	23	-	-
miR11490 *	TAGCCCGAACGCACAATTGGA	21	-	-
miR11532 *	TGACATTGTAAAGTACGGGAAT	22	-	-
miR3701 *	TAAACAATGTCCACCCTTCATT	22	-	-
miR397	TCATCATTGAGTGCAGCATTG	21	-	-
miR535	TCGACAACGAGAGAGAGCACGC	22	-	-
miR946 *	TCAGACCTTCTCCTATCCACAAT	23	-	-
miR946 *	CAGCCCTTCTCCTATCCACCAAC	23	-	-
miR946 *	TAGACCTTCTCCTATCCACCAAT	23	-	-
miR950 *	TCACATCTAGGCCACGATGGTT	22	-	-
miR950 *	TCACGTCTGGGCCTCTATGGTT	22	-	-
miR950 *	TCGTGTCCTCGGTGGTTTATGA	22	-	-
miR950 *	TTTACGTCTGGTCCTCGATGGTT	23	-	-
miR950 *	TTACATCTGGGCCACGGTGGTT	22	1	TC178188

* Conifer-specific conserved microRNAs.

**Table 2 genes-16-00026-t002:** Differentially expressed microRNAs with statistical significance *p*-value ≤0.05 (*P. sylvestris,* Taxus × media, *P. abies*) or ≤0.01 (*T. chinensis*). “–” represents miRNAs that were not differentially expressed.

microRNA Family	*Pinus sylvestris*	*Taxus chinensis*	Taxus × Media	*Picea abies*
miR156	-	down	down	-
miR159	up	-	-	-
miR162	-	-	-	down
miR164	up	up	up	-
miR167	-	-	-	down
miR168	-	down	down	-
miR169	up	down	down	up
miR172	-	down	down	up
miR319	-	-	-	down
miR390	-	up	up	up
miR396	up	down	down	down
miR397	down	-	-	up
miR398	up	-	-	up
miR398	-	-	-	up
miR399	-	-	-	-
miR408	up	-	down	-
miR437	-	-	-	up
miR480	-	down	-	-
miR482	-	-	-	up
miR529	-	-	-	down
miR535	down	-	-	-
miR858	-	-	-	up
miR946	down/up	-	-	-
miR947	up	-	-	-
miR950	down/up	-	-	up
miR1310	-	down	-	-
miR1311	-	-	-	up
miR1312	up	-	-	-
miR1313	up	-	-	down
miR1314	up	-	-	-
miR3693	up	-	-	up
miR3701	down	-	-	up
miR4376	-	-	-	up
miR6171	up	-	-	-
miR00002	up	-	-	-
miR00024	up	-	-	-
miR00028	up	-	-	-
miR11452	up	-	-	-
miR11466	down/up	-	-	-
miR11467	-	-	-	down
miR11487	up	-	-	up
miR11490	down	-	-	-
miR11498	-	-	-	down
miR11504	up	-	-	-
miR11511	up	-	-	-
miR11524	-	-	-	up
miR11532	down	-	-	-
miR11539	-	-	-	up
miR11544	up	-	-	-
miR11551	up	-	-	up

## Data Availability

Unique small RNA sequences were deposited to the SRA (Short Read Archive, NCBI) with the accession number PRJNA531446 (https://www.ncbi.nlm.nih.gov/sra/?term=PRJNA531446). Transcriptome sequences were deposited to the SRA with the accession number PRJNA1115068 (https://www.ncbi.nlm.nih.gov/sra/PRJNA1115068).

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
