# Peer review of "Differential microRNA and Target Gene Expression in Scots Pine (Pinus sylvestris L.) Needles in Response to Methyl Jasmonate Treatment"

_genes, 2024, doi:10.3390/genes16010026_

Round 1
Reviewer 1 Report
Comments and Suggestions for Authors
The author wrote in the M&M (Lines 92-94) that three Scots pine ramets where treaded with methyl jasmonate and other three like control. They extract RNA in two replicates (line 101-102) and six of them for transcriptome sequencing (line 110) and six for small RNAs sequencing (line 111).
In the SRA (Short Read Archive, NCBI) the accession PRJNA531446 contains 7 small RNAs samples and the accession PRJNA1115068 contains 8 RNA-seq samples (Lines 709). These number are different and it is not explained why, furthermore it is not clear which deposted samples are control and which are treaded with methyl jasmonate (e .g. sm5-1 small RNA library stand for what?). The date of sumbission to NCBI it is different 2019 (PRJNA531446) and 2024 (PRJNA1115068). Are the samples the same, RNA frozen for 5 years? This should be explained in the M&M.
The small RNAs data and analysis have been already presented in this article:
Krivmane, B.; Šņepste, I.; Škipars, V.; Yakovlev, I.; Fossdal, C.G.; Vivian-Smith, A.; Ruņgis, D. Identification and in Silico Characterization of Novel and Conserved MicroRNAs in Methyl Jasmonate-Stimulated Scots Pine (Pinus sylvestris L.) Needles. For. 2020, Vol. 11, Page 384 2020, 11, 384, doi:10.3390/F11040384.
It is not clear if the authors reanalyzed the data or what they use from the previous article.
The new data produced in this work are the RNA-seq samples (PRJNA1115068), but the analysis of these data is not presented well. Here some suggestions:
- in the M&M they said that they use the PS2_trinity trascriptome (Line 144) to map their RNA-seq samples. They should said how many RNA-seq reads are mapped and unmapped and how many genes are indentify in the transcriptome. Both for control and treatment.
- A differential expression (DE) analysis based only on the RNA-seq data (treatment Vs control) should be presented the number of up and down regulated genes.
- DE analysis of miRNA (treatment Vs control) should be presented the number of up and down.
- psRNATarget analysis between all 4 combinations ( RNA-seq upregulated & miRNA upregulated, RNA-seq downregulated & miRNA upregulated …)
- GO Analysis of the selected genes by psRNATarget analysis.
In conclusions it is not well explained the analysis done on the RNA-seq data and there is a big problem with the different samples’ number between the data deposited to NBCI and the M&M text.
Author Response
The author wrote in the M&M (Lines 92-94) that three Scots pine ramets where treaded with methyl jasmonate and other three like control. They extract RNA in two replicates (line 101-102) and six of them for transcriptome sequencing (line 110) and six for small RNAs sequencing (line 111).
In the SRA (Short Read Archive, NCBI) the accession PRJNA531446 contains 7 small RNAs samples and the accession PRJNA1115068 contains 8 RNA-seq samples (Lines 709). These number are different and it is not explained why, furthermore it is not clear which deposted samples are control and which are treaded with methyl jasmonate (e .g. sm5-1 small RNA library stand for what?). The date of sumbission to NCBI it is different 2019 (PRJNA531446) and 2024 (PRJNA1115068). Are the samples the same, RNA frozen for 5 years? This should be explained in the M&M.
The small RNAs data and analysis have been already presented in this article:
Krivmane, B.; Šņepste, I.; Škipars, V.; Yakovlev, I.; Fossdal, C.G.; Vivian-Smith, A.; Ruņgis, D. Identification and in Silico Characterization of Novel and Conserved MicroRNAs in Methyl Jasmonate-Stimulated Scots Pine (Pinus sylvestris L.) Needles. For. 2020, Vol. 11, Page 384 2020, 11, 384, doi:10.3390/F11040384.
It is not clear if the authors reanalyzed the data or what they use from the previous article.
The new data produced in this work are the RNA-seq samples (PRJNA1115068), but the analysis of these data is not presented well.
In conclusions it is not well explained the analysis done on the RNA-seq data and there is a big problem with the different samples’ number between the data deposited to NBCI and the M&M text.
Thank you for the suggested clarifications. The materials and methods section has been supplemented with additional information about the analysis of the data. In the previous article (Krivmane et al 2020), conserved and novel miRNAs from this data set and precursor sequences were identified, but in the present study, differentially expressed miRNAs were identified and analyzed. An analysis of the transcriptome and differentially regulated genes using the transcriptome data set has been reported previously (Kānberga-Siliņa, K.; Rauda, E.; Šķipars, V.; Vivian-smith, A.; Yakovlev, I.; Krivmane, B.; Šņepste, I.; Ruņģis, D. Transcriptomic response to methyl jasmonate treatment of Scots pine (Pinus sylvestris) seedlings. Environ. Exp. Biol. 2017, 257–274, doi:10.22364/eeb.15.27.), but in the present study, target genes of differentially expressed miRNAs were identified from the transcriptome data. Text has been added to the methods section to clarify this.
Regarding the discrepancies in the number of samples/libraries deposited in NCBI, some libraries were resequenced due to a low number of reads obtained in the initial sequencing run compared to the other libraries. Text has been added to the methods section to clarify this.
Responses to specific comments:
- in the M&M they said that they use the PS2_trinity trascriptome (Line 144) to map their RNA-seq samples. They should said how many RNA-seq reads are mapped and unmapped and how many genes are indentify in the transcriptome. Both for control and treatment.
- A differential expression (DE) analysis based only on the RNA-seq data (treatment Vs control) should be presented the number of up and down regulated genes.
As described above, the transcriptome analysis has been previously published, and in the present study, target genes of differentially expressed miRNAs were identified from the transcriptome data. Text has been added to the methods section to clarify this.
- DE analysis of miRNA (treatment Vs control) should be presented the number of up and down.
DE analysis of miRNAs (treatment vs control samples) is been presented at supplementary file 4, as well as in the manuscript (Line247). 43 up-regulated and 15 down-regulated miRNAs in MeJA treated samples compared to the control samples were identified The text has been edited to clarify this.
- psRNATarget analysis between all 4 combinations ( RNA-seq upregulated & miRNA upregulated, RNA-seq downregulated & miRNA upregulated)
This data has been included in Supplementary file 6
- GO Analysis of the selected genes by psRNATarget analysis.
This data has been included in Supplementary file 4 in the sheets Blast2GO_up and Blast2GO_down.
Reviewer 2 Report
Comments and Suggestions for Authors
Manuscript number: genes-337172
Manuscript title: Differential miRNA and target gene expression in Scots pine (Pinus sylvestris) needles in response to methyl jasmonate treatment
Authors: Baiba Krivmane and Dainis Edgars Ruņģis
The topic treated in the present manuscript is of potential interest for the Journal readership, in terms of novelty and impact. The amount of the experimental work done is remarkable, and the experimental methodology appears to be adequate. A valuable and detailed effort in done to put the results of the present experiments against the background of the available literature. The manuscript is well written and easy to follow, linguistically and conceptually speaking.
A single major point plus a number of minor ones need adequate consideration on the part of the Authors.
Major point:
1) Statistics: a formal paragraph describing the statistical approach adopted for evaluating the experimental results should be placed at the end of the M&M section.
Minor points:
2) Line #60: what the term “aos” stands for? Maybe “allene oxide synthase”?... Please clarify
3) The word “this” on lines # 83 and 86 should be replaced by “the present”, to avoid ambiguity
4) Please use the capital letter “L” for liter, and not the small one (e.g. on line #92).
5) EVERY uncommon acronym should be spelled out in full upon its FIRST mention in the text
6) Please avoid using acronyms in the figures’ legends/tables’ headings. Each legend/heading should be self-consistent and self-explanatory, without the need to refer to the text
7) Please avoid using acronyms in the paragraphs headings (e.g. line #400)
8) Please spell out in full linnean binomials upon their FIRST mention (e.g. lines # 370 and 371)
All the above considering, I recommend major revision of the present manuscript, properly and carefully addressing all the points raised above.
Author Response
The topic treated in the present manuscript is of potential interest for the Journal readership, in terms of novelty and impact. The amount of the experimental work done is remarkable, and the experimental methodology appears to be adequate. A valuable and detailed effort in done to put the results of the present experiments against the background of the available literature. The manuscript is well written and easy to follow, linguistically and conceptually speaking.
A single major point plus a number of minor ones need adequate consideration on the part of the Authors.
Major point:
1) Statistics: a formal paragraph describing the statistical approach adopted for evaluating the experimental results should be placed at the end of the M&M section.
A paragraph describing the statistical analyses and parameters has been added to the methods section
Minor points:
2) Line #60: what the term “aos” stands for? Maybe “allene oxide synthase”?... Please clarify
3) The word “this” on lines # 83 and 86 should be replaced by “the present”, to avoid ambiguity
4) Please use the capital letter “L” for liter, and not the small one (e.g. on line #92).
5) EVERY uncommon acronym should be spelled out in full upon its FIRST mention in the text
6) Please avoid using acronyms in the figures’ legends/tables’ headings. Each legend/heading should be self-consistent and self-explanatory, without the need to refer to the text
7) Please avoid using acronyms in the paragraphs headings (e.g. line #400)
8) Please spell out in full linnean binomials upon their FIRST mention (e.g. lines # 370 and 371)
All the minor points have been corrected in the revised manuscript. Acronyms have been spelled out in full upon first mention in the text, and replaced with full text in figures and tables. Linnean binomials have been corrected.
All the above considering, I recommend major revision of the present manuscript, properly and carefully addressing all the points raised above.
Round 2
Reviewer 1 Report
Comments and Suggestions for Authors
The quality of the manuscript has increased.
Reviewer 2 Report
Comments and Suggestions for Authors
Comments and criticisms raised on the original submission have been met by the Authors in a satisfactory manner